# Biocompatibility Assessment of Zinc Alloys as a New Potential Material for Bioabsorbable Implants for Osteosynthesis

**DOI:** 10.3390/ma16155224

**Published:** 2023-07-25

**Authors:** Maria Roesner, Sergej Zankovic, Adalbert Kovacs, Moritz Benner, Roland Barkhoff, Michael Seidenstuecker

**Affiliations:** 1G.E.R.N. Tissue Replacement, Regeneration & Neogenesis, Department of Orthopedics and Trauma Surgery, Medical Center-Albert-Ludwigs-University of Freiburg, Faculty of Medicine, Albert-Ludwigs-University of Freiburg, Hugstetter Straße 55, 79106 Freiburg, Germany; maria.roesner@uniklinik-freiburg.de (M.R.); sergej.zankovic@uniklinik-freiburg.de (S.Z.); 2Limedion GmbH, Coatings and Surface Analysis, Am Schäferstock 2-4, 68163 Mannheim, Germany; kovacs@limedion.de (A.K.); m.benner@limedion.de (M.B.); 3Quadralux e.K., Am Schäferstock 2-4, 68163 Mannheim, Germany; r.barkhoff@quadralux.de

**Keywords:** zinc alloy, zinc silver alloy, biocompatibility, osteosynthesis, bioabsorbable

## Abstract

In the last several years, zinc and its alloys have come into focus as bioabsorbable materials by qualifying themselves with an excellent corrosion rate, mechanical properties, anti-bacterial effects. and considerable biocompatibility. In this study, the biocompatibility of zinc–silver alloys containing 3.3 wt% silver (ZnAg3) was assessed by evaluating their cell viability, the proliferation rate, and the cell toxicity. Two alloys were investigated in which one was phosphated and the other was non-phosphated. The alloys were tested on human osteoblasts (hOb), which are, to a large extent, responsible for bone formation and healing processes. The performance of the phosphated alloy did not differ significantly from the non-phosphated alloy. The results showed a promising biocompatibility with hOb for both alloys equally in all conducted assays, qualifying ZnAg3 for further investigations such as in vivo studies.

## 1. Introduction

Orthopedics and trauma surgery are major fields in surgical medicine and globally up to 28.3 million surgical procedures were performed per year between 2017 and 2022 with an increasing tendency [1]. A great portion of these surgeries were conducted with hardware such as metal implants [1]. Hardware such as screws and nails are commonly used in surgery to reconnect fracture segments and stabilize the affected bone structure. The Global Burden of Disease (GBD) 2019 Fracture Collaborators [2] reported 455 million prevalent cases of fracture in 2019. For several years, there was still the question pending whether and under which circumstances an implant should be removed, especially after trauma surgery [3,4,5,6]. The decision yields significant medical consequences for the patient as well as economic consequences for the hospitals since the removal requests a second procedure, including all the risks and costs of a conventional surgical intervention [7]. With bioabsorbable implants entering the market, this question is redundant [8]. In the last decades, numerous studies have been conducted to investigate different alloys to find an ideal material for medical applications which would meet the high requirements in terms of mechanical properties and biocompatibility [8,9,10,11,12,13,14,15,16]. Bioabsorbable materials are expected to degrade at a speed adequate for the related tissue, providing the needed stability over time and allowing the natural healing process [7,11]. Until today, one of the first choices in orthopedics and trauma surgery for metal implant usage is titanium and its alloys [9], although long-term exposure can still bring the risk of foreign body reaction or inflammation [9,17]. Despite the good performance of a permanent implant, the issues on stress shielding of the organism and a possibly necessary second surgery remain [8].

Within the last years, zinc alloys have progressively gained the attention of the scientific community [14,15,16,18,19,20,21,22,23,24,25].Zinc is one of the trace elements of the human body and shows a great biocompatibility [16]. Studies by, Vojtěch et al. [14], for example, have demonstrated that the absorbed amounts of zinc coming from a degrading zinc implant are marginal compared to the tolerable daily dose of zinc, which is limited to a maximum intake of 40 mg/day [14,26]. When degrading, zinc is not known to convert to hydrogen, which can affect the healing process by creating potentially harmful hydrogen gas pockets in the surrounding tissue, which is a major issue with magnesium [23,27,28,29,30]. Furthermore, zinc is identified as an activator of aminoacyl-tRNA synthetase in osteoblastic cells, which leads to increased bone formation and mineralization. In addition to that, zinc has an inhibitory effect on osteoclastic bone resorption because it interferes with the osteoclast-like cell formation [31,32]. The degradation rate of the material is fundamental to its use as a bioabsorbable implant, whereby iron-based alloys are found to be below the benchmark of the recommended rate and magnesium-based alloys degrade too fast, while zinc alloys degrade at a reasonable rate [16,19,33,34]. Zinc and its alloys also meet the requirements regarding the corrosion rate as Hagelstein et al. [35] showed. In addition, Li et al. [36] has also shown that zinc–silver (ZnAg) alloys have antibacterial properties and exhibit no harmful toxicity [9,34,36].In addition to the promising biocompatibility, anti-inflammatory properties, and excellent corrosion rate, pure zinc itself seems to lack the necessary mechanical properties [19,21,37]. For this reason, alloying it with other materials could improve the mechanical properties and qualify it as an excellent biomaterial [19,21,32].

The aim of this study was to assess the biocompatibility of two ZnAg alloys, which differed in their surface by being either phosphated or non-phosphated. Mostaed et al. [34] described a series of ZnAg alloys with an Ag percentage between 2.5 and 7.0 wt% [21,34]. At 7.0 wt% Ag, the improvement of the mechanical properties regrading tensile strength reached their maximum at around 287 MPa [21]. Alloying zinc with silver tremendously improved the anti-bacterial properties of the material since silver is well known to inhibit the adherence of bacteria on surfaces [32,34,38,39,40,41]. Furthermore, silver is widely used in implant medicine and is proven to be highly qualified as a bone implant [38,42]. Previous studies have identified ZnAg alloys as promising biomaterials and showed great biocompatibility through in vitro as well as few in vivo studies [15,16,23,24,32,36,41,43]. As one of the most abundant minerals in the human body, phosphorus participates in vital processes [44]. Approximately 85% of phosphorus present in the body can be found in bone material, mostly building complexes with calcium [45]. The phosphatation of metallurgical samples in previous studies was shown to alter the corrosion rate [46] and might have further effects on the material and its properties. The objective of this study is to prove the expected biocompatibility of both alloys and to determine whether the phosphated coat has any effect on the outcome.

## 2. Materials and Methods

### 2.1. Sample Preparation and Characterization

The assessed alloys provided by Limedion GmbH, Mannheim, Germany, contained zinc and silver with an Ag contingent of approximately 3.3 wt% described as ZnAg3. An Indutherm VC 600 casting machine was used for casting the zinc–silver alloys. To reduce grain sizes and enhance mechanical properties, a cylindrical extrusion process at elevated temperatures (250–300 °C) was applied. During the extrusion process, sections of cast rod with a height between 20 and 50 cm and a diameter of 30 mm were pressed into 6 mm thick rods (extrusion ratio 25:1). The rods were cut into 1 mm thick disks. Thus, the samples used in this study were disks that measured 6 mm in diameter and 1 mm in thickness. The ZnAg3 alloy used in this study had a tensile strength of 227 MPa and an elongation of 44% according to the manufacturer data.

Two alloys, which differed in their surfaces, were tested: one alloy was phosphated and the other alloy was non-phosphated. As phosphating medium, an acidic zinc–calcium-based phosphate solution was applied, resulting in phosphate layers of a layer weight of 2.0 g/m^2^.

#### 2.1.1. Three-Dimensional Laser Scanning Microscopy

Both alloys were monitored throughout the experiment regarding the surface roughness. The roughness was determined at 200× magnification by using a 3D-laser scanning microscope (Keyence VK-X 200; Keyence, Osaka, Japan). For each alloy, 6 samples were evaluated and measured on 4 positions sectioned in 10 segments per sample. Pictures of the measured samples were taken at 100× magnification for accuracy and assembled digitally to obtain one bigger picture. The samples were not treated in any way before the measurements and imaging.

#### 2.1.2. Energy Dispersive X-ray (EDX) Spectroscopy and Environmental Scanning Electron Microscope (ESEM)

The EDX was performed to determine the exact ratio of zinc and silver contained in both alloys. ESEM pictures were taken of both alloys at 2554× and 10,217× magnification with a large field secondary electron detector and 20 kV acceleration voltage at approximately 10^−3^ Pa. Both were carried out on an ESEM (FEI Quanta 250 FEG, FEI, Hilsboro, OR, USA) equipped with an EDX system (Oxford Instruments INCA x-act, Oxford Instruments, Abingdon, Oxfordshire, UK).

#### 2.1.3. Inductively Coupled Plasma-Optical Emission Spectroscopy (ICP-OES)

To determine the metal concentrations of the ZnAg alloy, ICP-OES measurements were performed. Samples of about 0.1–0.2 g were cut out from three locations (bottom, middle, and top) of the 30 mm diameter cast ingot. After dilution in concentrated HNO3, working solutions were prepared by adding 2% HNO3. The measurements were carried out on a Perkin Elmer Avio 200 (Perkin Elmer, 710 Bridgeport Avenue, Shelton, CT 06484-4794, USA)

#### 2.1.4. Atomic Absorption Spectroscopy (AAS)

According to ISO standard 10993-15:2019-11 [47], 5 samples of each alloy were incubated in Eppendorf tubes 2.0 mL containing 1 mL Tris buffer (TRIS Hydrochlorid, 1 kg Pufferan^®^ ≥99%, p.a., article No. 9090.3, Roth^®^, Karlsruhe, Germany) for 10 days at room temperature. The pH value of the Tris buffer was set to 7.4 with hydrochloric acid solution (hydrochloric acid solution, Volumetric, Reag. Ph. Eur., 0.1 M HCl (0.1 N), Honeywell, Charlotte, NC, USA). After 1, 2, 3, 7, and 10 days, the 1 mL Tris buffer was transferred to an empty Eppendorf tube 2.0 mL and replaced with new Tris buffer. The collected solutions per day and sample were analyzed separately at the Institute for Geoscience, University of Freiburg, Germany, with an atomic absorption spectrometer (Perkin Elmer AAS 4110ZL Zeeman, Perkins Elmer, Waltham, MA, USA).

### 2.2. Biocompatibility

The evaluation of the biocompatibility was conducted according to ISO standard 10993-1:2021-05 [48]. Prior to usage in the cell culture experiments, the samples were sterilized following a specific protocol to avoid uncontrolled oxidation. The samples were separately immersed in 1 mL of different solutions in 2.0 mL Eppendorf tubes and sterilized in the ultrasonic bath (Branson Bransonic ultrasonic cleanser, 5510E-DTH, Branson, Danbury, CT, USA) for a defined amount of time. First, the samples were sterilized in n-hexan (n-Hexan, Rotipuran^®^ ≥ 99%, p.a., article No. 4723.2, Roth^®^, Karlsruhe, Germany) for 20 min in the ultrasonic bath. After replacing the n-hexan with acetone (Acetone, Rotipuran^®^ ≥ 99.8%, p.a., article No. 9372.1, Roth^®^, Karlsruhe, Germany), they were sterilized for another 20 min followed by a 3 min sterilization with 100% ethanol (ethanol absolut ≥ 99.8%, AnalaR Normapur^®^ ACS, Reag. Ph. Eur. analytisches Reagens, VWR International, Darmstadt, Germany) and another 20 min with 70% ethanol. Finally, the samples were transferred separately into 2.0 mL Eppendorf tubes containing 250 μL medium (Gibco^®^ Medium 199 (1X), article No. 31150-022, Gibco^®^, Grand Island, NE, USA; additives: 10% FBS and 1% P/S) and incubated for 24 h at 37 °C and a 5% CO_2_ saturation in an incubator (CO_2_ incubator ICO50, Memmert, Schwabach, Germany) to reach an homogeneous oxidated surface in all samples since oxidation naturally cannot be avoided neither in the cell culture nor in any organism. Furthermore, this step was needed to produce the eluate for the eluent trials.

Human osteoblasts (hOb) were used for all cell culture experiments and were obtained with informed consent and isolated from bone material obtained through surgical therapy such as total knee arthroplasty (Ethics vote FREEZE 418/19 of the ethics commission of Freiburg University Medical Center). All biocompatibility assays were conducted with 10,000 cells/40 μL per sample. For all assays, we tested a triplet of each alloy in direct contact trials and in eluent trials with the dilutions 1 to 6, 1 to 10, and 1 to 15. Three runs per assay were performed. The eluates of all individual samples were collected, and the different dilutions were prepared with the needed amount of eluate and medium. It should be noted that the calculations of the eluate for the cell proliferation assay (WST assay) and cytotoxicity assay (LDH assay) had to consider the added 40 μL medium containing the cells to avoid adulterations of the dilutions. After 3 and 7 days, a medium change as well as a change of the eluates was carried out.

#### 2.2.1. Live/Dead Assay

For the live/dead assay, 40 μL of medium with a concentration of 250,000 cells/mL was pipetted onto each sample. The eluent trials were performed on Thermanox^®^ membranes (Nunc^®^, Thermanox^®^ Plastic Coverslips, catalog No. 174950, Nunc Bran Products, Rochester, NY, USA) since the samples needed to be transferable for evaluating them under the microscope later on. Due to the size of the Thermanox^®^ membranes, the live/dead assay was conducted on 24-well plates. The samples were incubated for 2 h at 37 °C and a 5% CO_2_ saturation in the incubator (Midi CO_2_-incubator, catalog No. 3403, Unity Lab Services as part of Thermo Fisher Scientific, Waltham, MA, USA). Then, 1 mL medium and 1 mL of the needed eluate, respectively, was added to each well and the well plates were incubated for 3, 7, or 10 days. For each of the different durations of time, a separate well plate was pipetted. After the incubation, the samples were evaluated by staining them with a live/dead cell staining kit (Viability/Cytotoxicity Assay Kit for Animal Live & Dead Cell, catalog No.: 30002-T, Biotium, Fremont, CA, USA). The entire staining process, as well as the preparation of the staining solution, was performed in the dark to prevent photobleaching. A total of 2 mL of Dulbecco’s Balanced Salt Solution (DPBS (1X), article No. 14190-094, Gibco^®^, Grand Island, NE, USA) was pipetted into a Falcon tube and 4 μL of ethidium homodimer III (Eth D-III) solution was added. Then, the solution was vortexed (Corning^®^ LSE Vortex Mixer, catalog No. 6776, Corning, NY, USA) for a few seconds on a low level and 1 μL calcein was added. The samples were washed with DPBS and stained for 10 min in the prepared staining solution. The evaluation was conducted with an Olympus fluorescence microscope (BX51, Olympus, Osaka, Japan) at 5× and 10× magnification. For each sample, 5 pictures were taken, 1 overview and 4 detail shots. The calculations were performed with the counted cells of the detail shots at 10× magnification.

#### 2.2.2. Cell Proliferation (WST Assay)

As in the previously described biocompatibility test, the cells were pipetted onto the samples and incubated. The WST assay was conducted on 48-well plates for the direct contact trials and 96-well plates for the eluent trials since the smaller sized working volume did not require as much eluate. Furthermore, each well plate comprised one triplet of blank wells only containing medium and one triplet of control wells containing medium with cells. After 2 h of incubation, 460 μL medium was added to each well on the 48-well plates and 160 μL of medium and 160 μL of the needed eluate, respectively, was added to each well on the 96-well plate. The well plates were incubated for 1, 3, or 7 days. A separate well plate was prepared for each duration of time. After the incubation, the medium or eluate was removed and the wells were washed three times with DPBS. For the direct contact trials, the samples were transferred to a new well on the same well plate to evaluate just the cells being located on the sample. A total of 500 μL medium without phenol red (Gibco^®^ Medium 199 (1X), article No. 11043-023, Gibco^®^, Grand Island, NE, USA; additives: 1% FBS and 1% P/S) was pipetted into the wells containing a sample and 300 μL medium without phenol red was pipetted into the rest of the wells. For the eluent trials, 200 μL medium without phenol red was pipetted into each well. In total, 10% of the WST solution (Cell Proliferation Reagent WST-1, article-No. 11644807001, Roche, Basel, Switzerland) was added to each well and the plate was incubated for 2 h at 37 °C and a CO_2_ 5% saturation. Then, 3 × 100 μL of each well of the direct contact trials and 2 × 100 μL of each well of the eluent trials were transferred to a new 96-well plate. All steps involving the WST solution were performed in the dark. Finally, the absorbance was measured at 450 nm using the Spectrostar nano (BMGlabtech, Ortenberg, Germany) spectrometer.

#### 2.2.3. Cytotoxicity (LDH Assay)

The LDH assay was entirely performed with medium without phenol red since the medium was not replaced at any time and was directly evaluated by measuring the absorbance. According to the previous scheme, the same amount of cells were pipetted onto the samples and incubated. The LDH assay was conducted on 48-well plates for the direct contact trials and 96-well plates for the eluent trials and comprised of one triplet of blank wells alongside one triplet of negative and positive control wells. The negative control contained the cells and medium and was described as 0% toxicity. The positive control contained the cells and a solution consisting of medium with 1% Triton X and was described as 100% toxicity. After 2 h of incubation, 460 μL medium or 460 μL of the Triton X solution was added to each well on the 48-well plates and 160 μL medium, 160 μL of the Triton X solution or 160 μL of the needed eluate was added to each well on the 96-well plates. The well plates were incubated for 1, 2, or 3 days. A separate well plate was prepared for each duration of time. Then, 3 × 100 μL of each well of the direct contact trials and 100 μL of each well of the eluent trials was transferred to a new 96-well plate. The used cytotoxicity detection kit (Cytotoxicity Detection Kit (LDH), article. No. 11644793001, Roche, Basel, Switzerland) contained a catalyst solution which had to be added in a 1 to 45 ratio to a staining solution. Therefore, 160 μL of the catalyst solution was pipetted into 7.2 mL of the staining solution. Then, 100 μL of this solution was added to each well and the well plates were incubated for 30 min at room temperature. All steps involving the staining solution were performed in the dark and the well plates were sheltered in aluminum foil and placed in a dark room for the time of incubation. Finally, the absorbance was measured at 490 nm using the Spectrostar nano (BMGlabtech, Ortenberg, Germany) spectrometer. The calculations were performed according to the product information sheet [49].

#### 2.2.4. pH-Measurements

The pH value was monitored throughout the cell culture experiments with a pH meter (Laboratory meter inoLab pH 7110 SET 4 pH/mV meter, article-No. WW1AA114, WTW as part of xylem brand, Washington, DC, USA). The medium that came into contact with the samples was collected during the medium changes. Thus, we could measure the pH value during the incubation with cells after 3, 7, and 10 days. For comparison, pure medium with and without phenol red as well as excessive eluate directly after production was also measured. The measurements on day 3, 7, and 10 were conducted with medium with phenol red.

### 2.3. Statistics

All calculations and designs of diagrams were executed with Origin 2022 Professional SR1 (OriginLab, Northampton, MA, USA). The conducted biocompatibility tests are presented by their means and standard deviations.

## 3. Results

### 3.1. Sample Characterization

#### 3.1.1. Grain Size Determination

The following two microphotographs in Figure 1 show metallographic examinations of the alloy after casting (a) and after extrusion (b). The grain sizes of the cast alloy were in the range between 70 and 100 µm, while extrusion reduced the grain sizes to 5–10 µm (c).

#### 3.1.2. Three-Dimensional Laser Scanning Microscopy

The results of the surface roughness measurements showed an average roughness (Sa) with 0.52 ± 0.11 µm for the non-phosphated samples and 0.57 ± 0.10 µm for the phosphated samples (see Figure 2).

#### 3.1.3. Energy Dispersive X-ray (EDX) Spectroscopy

The EDX spectroscopy revealed that the suspected elements were found in both samples (Table 1 and Table 2). The carbon and oxygen found were unavoidable impurities on the surface as the EDX was not performed under sterile conditions and was rendered in the evaluations (Figure 3). The quantitative analysis showed an Ag weight% of 3.93 for the non-phosphated alloy, which is a slightly higher than expected (Table 1). For the phosphated alloy, the Ag weight% regarding just the zinc and silver would be 3.37 wt% if it was not for the phosphorus and oxygen (Table 1).

#### 3.1.4. Environmental Scanning Electron Microscope (ESEM)

For both alloys, the surface structure was already observed at 40 µm. Notably the structure of the non-phosphated alloy (Figure 4a,c) showed the presence of grind marks that occurred through the manufacturing process of the material. In the images of the phosphated alloy (Figure 4b,d), the crystal structures of the phosphatation are visible.

#### 3.1.5. Inductively Coupled Plasma-Optical Emission Spectroscopy (ICP-OES)

The ICP-OES measurement resulted in a value of 13.82 ± 2.44 mg/L Zn, corresponding to 96.7 ± 0.05% Zn. The specification of 96.7% Zn in the alloy was thus met. In addition, 0.47 ± 0.09 mg/L Ag was measured, corresponding to 3.30 ± 0.05% Ag. Here too, the specification of 3.3% silver in the alloy was met.


**Element**

**Concentration mg/L**

**Percentage %**

**Index Value %**
Zn13.82 ± 2.4496.70 ± 0.0596.70Ag0.47 ± 0.093.30 ± 0.053.30

#### 3.1.6. Atomic Absorption Spectroscopy (AAS)

The AAS did not show any significant difference in the measurements between the phosphated and the non-phosphated zinc alloys (Figure 5). Both alloys relieved atomic components increasingly until day 7. The silver was frequently in a non-measurable range. The phosphated alloy released slightly more silver components than the non-phosphated alloy.

### 3.2. Biocompatibility

#### 3.2.1. Live/Dead Assay

The number of living cells/mm^2^ in all the eluent trials was similar for all days and dilutions with 220 ± 138 cells/mm^2^ for the non-phosphated alloy and 366 ± 190 cells/mm^2^ for the phosphated alloy (see Figure 6). The direct contact trials were presented by clearly less cells with 15 ± 19 cells/mm^2^ for the non-phosphated alloy and 14 ± 22 cells/mm^2^ for the phosphated alloy (Figure 7 and Figure 8). The results are presented as percentages of living, unhealthy, and dead cells. Unhealthy cells are described as cells being not clearly identifiable as living cells, lacking sign of certain death which would be a red dyed cell nucleus. Unhealthy cells were not added to the counted cells/mm^2^. The dilution 1 to 15 (Figure 6a) showed the best cell viability for both alloys with a living cell percentage constantly above 94%. For the dilution 1 to 10 and 1 to 6 (Figure 6b,c), the results showed a non-significant tendency for a decrease in living cells. The statistics for the dilution 1 to 6 (Figure 6c) displayed percentages of living cells frequently below 90% but above 80%. The direct contact trials resulted in poorer cell viability (Figure 6d) with about 20–30% of living cells and a higher amount of unhealthy cells with percentages ranging between 40% and 60%. It is worth mentioning that the percentage of dead cells of the direct contact trials decreased significantly over time with 30.5% ± 21.6% on day 3, 16.2% ± 19.4% on day 7, and 10.4% ± 8.3% on day 10 for the non-phosphated alloy and 23.7% ± 21.7% on day 3, 14.2% ± 13.3% on day 7, and 10.8% ± 6.9% on day 10 for the phosphated alloy. The corresponding diagram can be found in Figure A1.

#### 3.2.2. Cell Proliferation (WST Assay)

The cell proliferation over 1, 3, and 7 days was assessed with the WST assay. The measured values of the spectrometer were converted into factors, with day 1 representing the starting point. The dilutions 1 to 15 and 1 to 10 showed in both alloys a positive proliferation tendency, although without significance. The dilution 1 to 6 showed different results in the two alloys. For the phosphated alloy, there was a clear positive proliferation tendency, and for the non-phosphated alloy, the proliferation rate was markedly poorer. For both alloys, the direct contact trials showed the slowest proliferation rate (Figure 9).

#### 3.2.3. Cytotoxicity (LDH Assay)

As Figure 10 shows, both alloys started with about 25% toxicity in all trials and reached about 0% toxicity within the second day. On the third day, neither of the two alloys showed toxicity in any trial.

#### 3.2.4. pH-Measurements

The measured pH values are listed in Table 3 for the non-phosphated and phosphated alloy. For the medium with phenol red, the pH value was 7.69, and for the medium without phenol red, the pH value was 7.76.

## 4. Discussion

### 4.1. Sample Characterization

The surface roughness did not differ much between both alloys, with an average roughness (Sa) of 0.52 µm ± 0.11 µm for the non-phosphated samples and 0.57 µm ± 0.10 µm for the phosphated samples, and showed that the results were not impacted by a possible difference in the attachment of the cells to the material due to the surface roughness. Furthermore, the measurements are in a similar range compared to previous studies on biomaterials [50,51]. Wątroba et al. hypothesized that a surface roughness of 0.22 µm is adequate for MG-63 cells [52]. According to Andrukhov et al. [53], even a surface roughness of 1–2 µm is suitable for osteoblastic cells.

The EDX detected a reasonable amount of phosphorus and oxygen in the phosphated alloy. It needs to be considered that the phosphated coat could possibly alter the result of the EDX for the phosphated alloy concerning the Ag wt%. However, the EDX results certainly depict the surface to which the cells were exposed during the cell culture experiments. In addition, the ESEM pictures represented a surface structure of the non-phosphated alloy similar to previous studies for ZnAg alloys [36,41,50]. The surface of the phosphated alloy revealed a crystal-like structure due to the phosphatation. These structures were also observed by Ashassi-Sorkhabi et al. [46].

The atomic absorption spectroscopy (AAS) showed that the released amount of zinc and silver was similar for both alloys. The measurements also revealed that Ag frequently was under the detection limit, which matches the results of Li et al. [41]. The low solubility of Ag in chloride solution might explain the results, although Ag is known to build soluble AgCl^2−^ complexes and, thus, can be present in the samples even in low concentrations. After day 7, a decrease in the zinc and silver concentrations occurred, which could be explained by a de-alloying effect on the surface. On the other hand, Zn(OH)_2_ and ZnO might have been formed and precipitated on the surface, causing the concentration of zinc to decrease before reaching a stable state as Li et al. [41] observed. Zinc was in a range of 50 to 230 µg/mL compared to the Li et al. [41] findings of a zinc ion concentration of 738.9 μM using DMEM + 10% FBS and 313.7 μM using McCoy’s 5A + 15% FBS. Furthermore, our results do not support the level of toxicity being determined by the zinc ion concentration in extracts for osteoblastic cells. Here, Kubásek et al. [54] proposed a maximum of zinc ion concentration of 120 µm for U-2 OS osteoblasts as the cellular tolerance limit.

### 4.2. Biocompatibility

The live/dead assay revealed a very high biocompatibility of both ZnAg3 alloys towards the human osteoblastic cells whereas all eluent trials showed the fraction of living cells to be above 80% throughout all dilutions and for every duration of time. In comparison, the direct contact trials resulted in a higher portion of unhealthy and dead cells. However, the amount of dead cells in the direct contact trials decreased significantly over time, as shown in Figure A1. Li et al. [41] evaluated the cell viability of a Zn–Ag–Au–V alloy with L929 cells and Saos-2 cells and displayed similar results for eluent trials. Unfortunately, they did not include the numbers of cells in the samples, but only the relative metabolic activity and the relative cell proliferation.

The WST assay showed no significant cell proliferation, however, human osteoblasts are known to have a very slow proliferation rate since it takes up 6 weeks for human osteoblasts after passaging in cell culture to gain a confluent plate [55]. Therefore, the results of the WST assay demonstrated that the number of living cells over time was maintained and the relative proliferation rate of the cells must not be altered necessarily. Holthaus et al. [56] also observed a very steady increase of cell proliferation for human osteoblasts over 7 days for titanium alloys.

The LDH assay showed a decreasing toxicity over 3 days for all trials and even reached a negative toxicity compared to the control on day 3. This emphasizes our assumption that ZnAg3 is well tolerated by human osteoblasts. Bosetti et al. [57] also displayed similar results for stainless steel- and silver-coated samples using the LDH and, therefore, showed that Ag does not exhibit any cytotoxicity and is well qualified as a biomaterial in orthopedics and trauma surgery. Wątroba et al. [52] also conducted the LDH assay on MG-63 with ZnAg3 among other materials and found higher concentrations of zinc ions to be cytotoxic. Our results did not show these issues in any of our runs of the LDH assay, while the direct contact trial had the highest zinc ion concentration and yielded no cell toxicity on day 3 for the non-phosphated alloy and was mostly comparable ZnAg3 alloy.

Since a material is considered cytotoxic if the reduction of the cell viability is greater than 30% according to ISO 10993 standard [48], the results do not provide any evidence of cytotoxicity. Collectively, the results of the live/dead assay, WST assay, and LDH assay independently suggest a consistent biocompatibility.

The measured pH values were similar for both alloys and reflect the increase of the pH value as the zinc ions of the material are relieved, which has also been proven in previous studies [41]. The physiological pH value of 7.36 should be maintained within a certain range [58]. During our trials, we found a pH value slightly higher than targeted. However, other materials increase the pH value even more drastically, such as magnesium alloys [59]. The human body provides more effective buffering systems than the cell culture which is a rather simulated environment for the cells.

## 5. Conclusions

Both zinc alloys demonstrated promising biocompatibility, which could be displayed in all performed assays in the eluent trials as well as in the direct contact trials. Considering the excellent mechanical properties, corrosion rate, and biocompatibility, zinc alloys are highly qualified for further investigations and future clinical use as a bioabsorbable implant. We could achieve consistent results throughout the study and disprove any possible cytotoxicity of ZnAg3 against human osteoblasts in vitro. As depicted in the literature, zinc and its alloys meet the high requirements set for bioabsorbable implants and seem to be distinguished candidates for biomaterials. We could not identify any differences in the results between the phosphate and non-phosphated alloy, which may be due to the low layer thickness. Prospectively, we aim to assess the aptitude of ZnAg3 in vivo for further evaluation.

## 6. Patents

The alloy, including its casting and extrusion process, is patented by the company Limedion. It includes the production of zinc with 90.0–99.5 wt% and silver with 0.05–10 wt%. The patent specification number is EP 3 250 247 B1.

## Figures and Tables

**Figure 1 materials-16-05224-f001:**
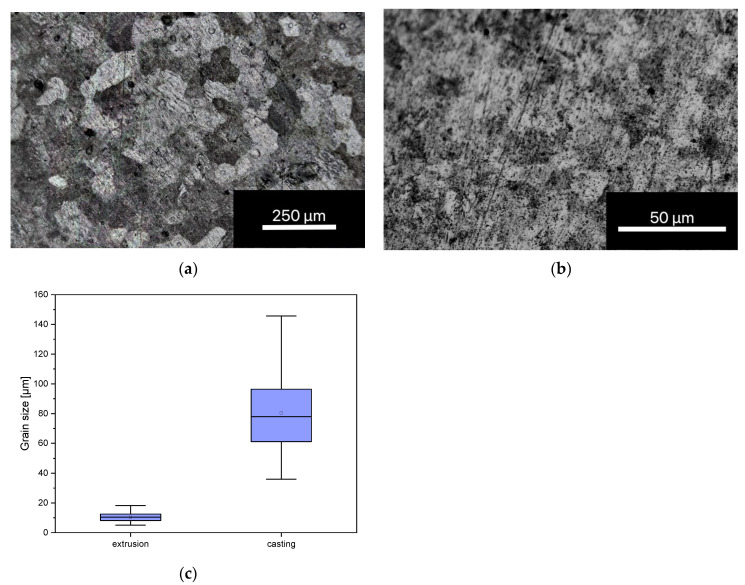
Photomicrographs of the grain size of the ZnAg3 alloy directly after casting, 20× magnification (**a**) and extrusion section of 50× magnification (**b**) and boxplot of the grain size after extrusion and casting (**c**). The photomicrographs were taken with a Leica DMRX 020-525.706 microscope. N = 30.

**Figure 2 materials-16-05224-f002:**
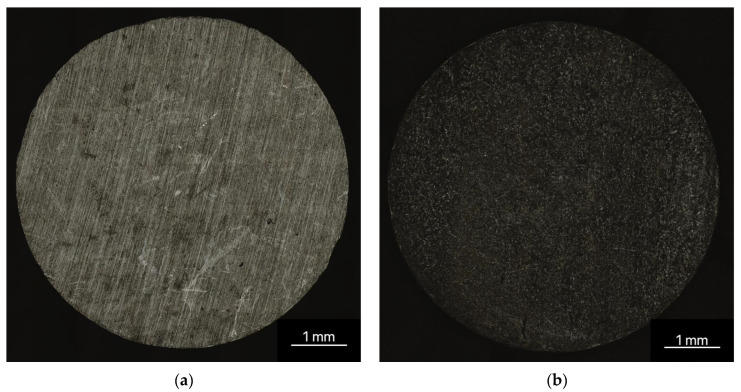
3D-laser scanning microscopic pictures of untreated non-phosphated (**a**) and phosphated (**b**) ZnAg3; images taken with the 3D-laser scanning microscope Keyence VK-X 200.

**Figure 3 materials-16-05224-f003:**
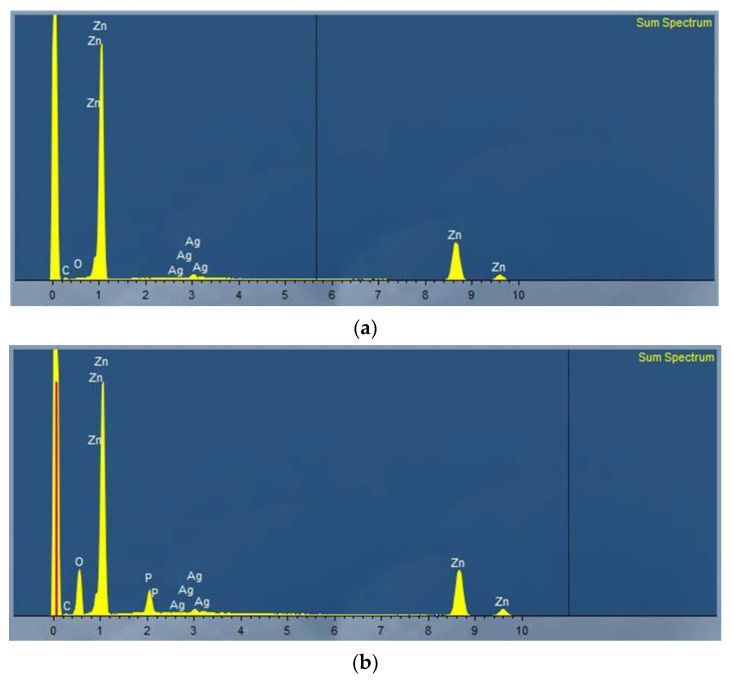
Sum spectrum (EDX) of untreated non-phosphated (**a**) and phosphated (**b**) ZnAg3.

**Figure 4 materials-16-05224-f004:**
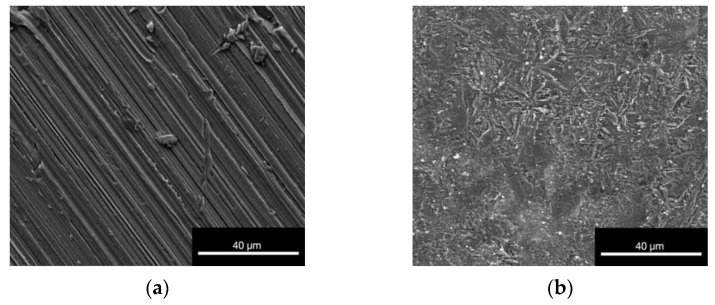
ESEM pictures of untreated non-phosphated (**a**,**c**) and phosphated (**b**,**d**) ZnAg3 at increasing magnifications; images taken with an FEI Quanta FEG 250 ESEM with large field secondary electron detector and 20 kV acceleration voltage at 10^−3^ Pa.

**Figure 5 materials-16-05224-f005:**
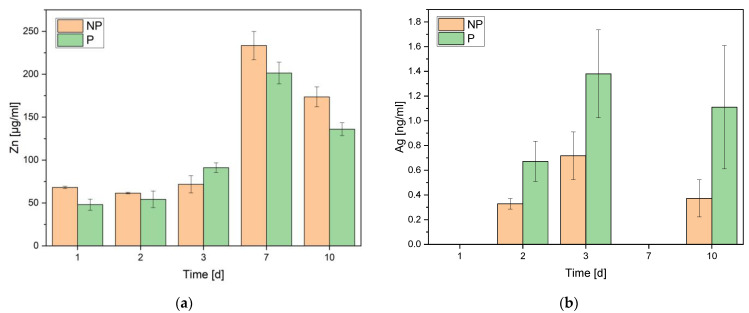
Atomic absorption spectroscopy of zinc (**a**) and silver (**b**) over 10 days of untreated non-phosphated (NP) and phosphated (P) ZnAg3.

**Figure 6 materials-16-05224-f006:**
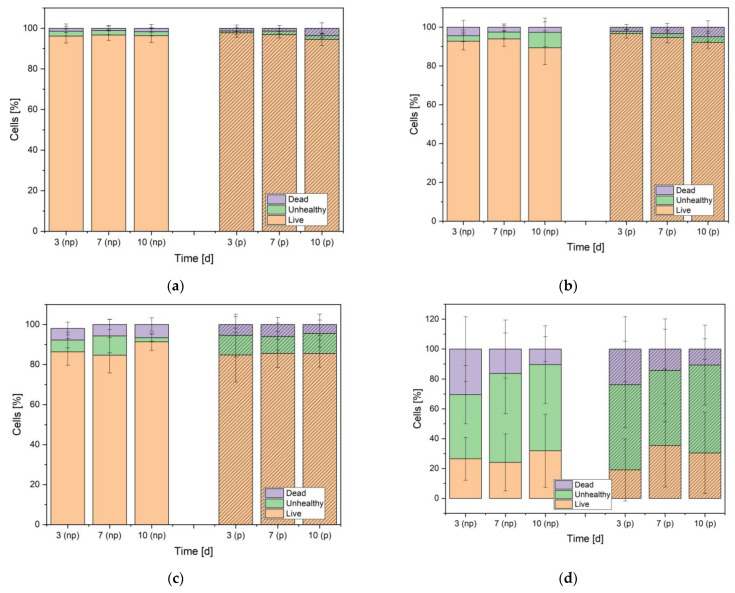
Live/dead assay showing living, unhealthy, and dead cells [%] after 3, 7, and 10 days for the dilution 1 to 15 (**a**), 1 to 10 (**b**), 1 to 6 (**c**), and for the direct contact trial (**d**) for non-phosphated (plain) and phosphated (shaded) ZnAg3.

**Figure 7 materials-16-05224-f007:**
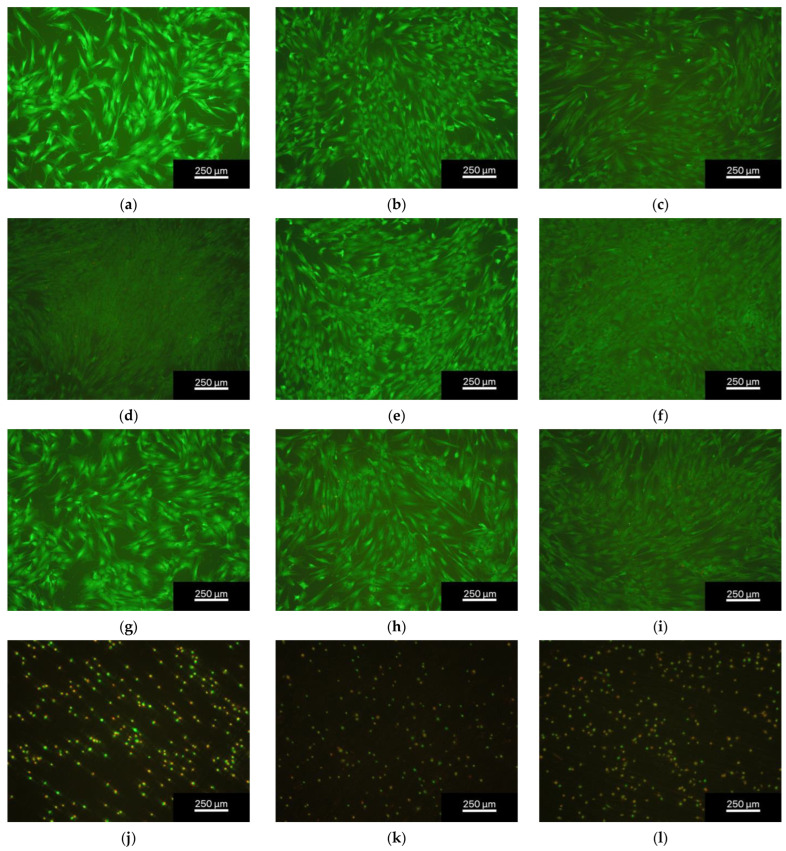
Fluorescence microscopic images of the non-phosphated ZnAg3 at 5× magnification, showing dead cells (red) and living cells (green); from left to right: days 3, 7, and 10 days of dilution 1 to 15 (**a**–**c**), 1 to 10 (**d**–**f**), 1 to 6 (**g**–**i**), and direct contact trial (**j**–**l**); images taken with an Olympus BX-53 fluorescence microscope.

**Figure 8 materials-16-05224-f008:**
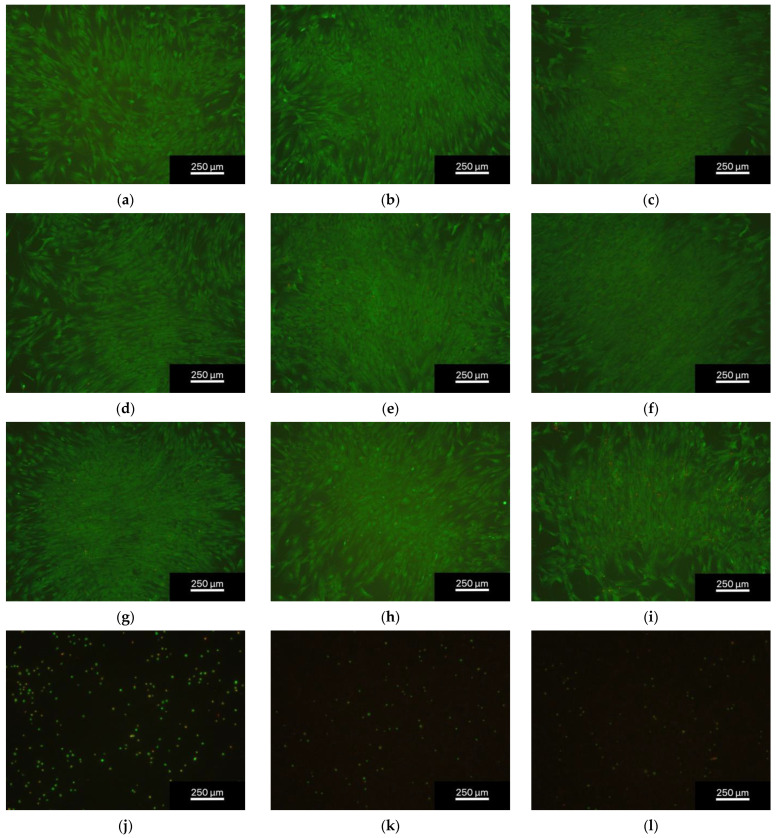
Fluorescence microscopic images of the phosphated ZnAg3 at 5× magnification, showing dead cells (red) and living cells (green); from left to right: days 3, 7, and 10 days of dilution 1 to 15 (**a**–**c**), 1 to 10 (**d**–**f**), 1 to 6 (**g**–**i**), and direct contact trial (**j**–**l**); images taken with an Olympus BX-53 fluorescence microscope.

**Figure 9 materials-16-05224-f009:**
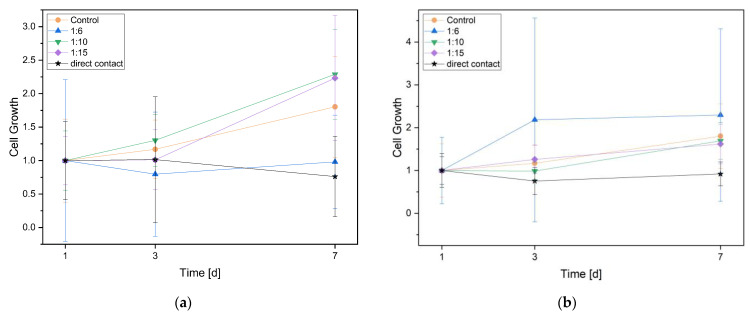
Cell proliferation on day 1, 3, and 7 for non-phosphated (**a**) and phosphated (**b**) ZnAg3. Medium with cells served as a control.

**Figure 10 materials-16-05224-f010:**
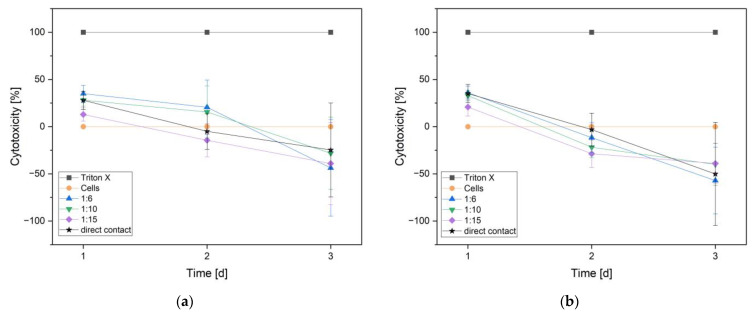
Cytotoxicity on day 1, 2, and 3 for non-phosphated (**a**) and phosphated (**b**) ZnAg3. A 1% triton X solution was defined as 100% cytotoxicity and, therefore, the positive control (Triton X), medium with cells served as a negative control defined as 0% toxicity (Cells).

**Table 1 materials-16-05224-t001:** Quantitative analysis of elemental spectrum (EDX) of untreated non-phosphated (np) and phosphated (p) ZnAg3 regarding zinc and silver.

Element	Weight% np	Weight% p
Zn	96.07 ± 0.22	96.63 ± 0.52
Ag	3.93 ± 0.22	3.37 ± 0.23

**Table 2 materials-16-05224-t002:** Quantitative analysis of elemental spectrum (EDX) of untreated phosphated ZnAg3 including phosphorus and oxygen.

Element	Weight%	Atomic%
Zn	67.02 ± 0.36	36.77
O	24.91 ± 0.34	55.82
P	5.73 ± 0.13	6.64
Ag	2.34 ± 0.16	0.78

**Table 3 materials-16-05224-t003:** pH values for non-phosphated ZnAg3 (np) and phosphated ZnAg3 (p) right after the production of the eluate and after 3, 7, and 10 days. The pH values were measured with Laboratory meter inoLab pH 7110 SET 4 pH/mV meter, WTW as part of xylem brand, Washington, DC, USA.

	np	p
Eluate (medium with phenol red)	8.48	8.50
Eluate (medium without phenol red)	8.69	8.64
Day 3	7.77	7.87
Day 7	7.76	7.76
Day 10	8.09	8.11

## Data Availability

The data presented in this study are available on request from the corresponding author.

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
