# Peer review of "Biocompatibility Assessment of Zinc Alloys as a New Potential Material for Bioabsorbable Implants for Osteosynthesis"

_materials, 2023, doi:10.3390/ma16155224_

Round 1
Reviewer 1 Report
The Manuscript is interesting, concerned on an important and contemporary topic. The research is based on several modern analytical techniques and provides a sufficient amount of high-quality data. My opinion is that this research should be presented in Materials as the high ranked journal.
However, a numerous flaws are present in the manuscript. Materials and Methods section is sometimes confusing and without precise data. It gives the impression that it was written hastily or compiled from earlier works. Some of the data seems to be wrong or at least need to be explained. In the Result section a lot of methods were used but not all of them were conclusive. Some figures are not clear and two of them have a wrong (mixed) captions.
Further, there are many typographic and grammar errors; several technical errors are also present in the text. I will point out the fact that I am not a native English speaker; however, I was able to identify a large number of linguistic errors (singular/plural forms, definite/indefinite articles, verb tenses, adjectives, prepositions, etc.). Nevertheless, I understood the meaning of phrases and sentences well (English is not that bad, but the text was not checked). Some (but not all) comments of this type are given further. Authors are kindly asked to check the text and fix the errors carefully. Proofreading of the text is recommended.
In conclusion, my recommendation is to accept the Manuscript with major revisions.
General Comments
A comparison of tensile strength between casted and extruded samples is missing. For the comprehensive material characterization given in the paper, samples' tensile strength and hardness should be included.
2. Materials and Methods
2.1. Sample Preparation and Characterization
This part (2.1.) is a little confusing and not easy to follow. It is not clear the importance of the dimensions of the alloy samples obtained from the supplier (Limedion GmbH) since it was melted and cast. More important is to give the dimensions of the casted sample, and that information is missing. Both dimensions should be shown (without explanation for the first; otherwise, it needs to be clarified). Authors also can provide just dimensions after extrusion. However, in 2.1.3. diameter of the cast ingot was given (30 mm), which is better to be in 2.1.
Line 92: It is not clear for the Ag content whether it is to producers' specifications or by analysis in the study.
Line 95: elevated temperatures (for the extrusion process) is not a precise term. Please give a precise temperature (or temperatures) for the operation.
Line 97: The phosphating procedure is not given!
2.1.2. Low vacuum has been used. Is this information just given by mistake? If not, a short explanation for an unusual procedure should be shown.
3. Results
3.1. Sample Characterization
3.1.1 Grain Size Determination
Line 266: Authors gave the same magnifications (as common practice), but here it is not beneficial. Figure (b) should be shown with 10x higher magnification, or part of the image should be enlarged; otherwise, the grains are not visible at all.
Line 269: The statistical sample is small for the purpose (average grain size). Fortunately, it is given approximatively so it can stay*.
*Standards for the number of grains per mm2 (and mean grain size) and software from Leica for the purpose, of performing the task on hundreds and thousands of grains.
Line 294: Are these EDX spectra necessary? They illustrate differences between two samples, but if authors consider them significant, then a short explanation is needed (tables 1 and 2 seem sufficient).
Line 300: Figures at 160x and 40868x magnifications (a, b, g and h) are not illustrative for this conclusion. If authors insist on them, it should be noted that it is better visible in figures c-d and e-f
Lines 303 and 304: Equipment and the parameters of measurement have already been given in the experimental part, and there is no need to repeat.
Lines 305-310 (the whole 3.1.4. part): ICP-OES just confirm the specifications from the producer of the alloy, and this could be moved to the experimental part. It looks superfluous (as a section in the Results part).
Line 315 (whole 3.1.5. part): Comment (short explanation) for the decrease in the concentration of metals is needed. For the zinc, it could be an extremely dealloying effect (selective dissolution of zinc) that leads to the higher silver concentration at the surface (but this need to be confirmed by EDS or some other analysis).
Although a highly sensitive graphite furnace AAS device was used, silver in chloride solutions is so poorly soluble that it was expected to be at the detection limit. On the other hand, in 0.1M chloride solution silver can exist as soluble in the form of AgCl2- ions in such extremely low concentrations (ng/ml, pg/L). Here one could even expect an increase in concentration with time, although that is more for a separate study and the use of ICP-MS. Here the results for silver are, in fact, inconclusive and maybe it can be stressed. Additionally, an increase in pH also can be the cause for the decrease in Zn2+ ion concentration, but here the buffer is used.
Line 327: Are values permuted here (15 ± 19 and 14 ± 22)? (the deviations are greater than the values)
Lines 366 and 375: Captions for Figures 9 and 10 are given incorrectly. Figure 9 has a caption for Fig. 10 and vice versa.
Line 380: (just a positive comment, do not change anything, it is for the disscusion part*). Important result! This is, fortunately, under a pH of ~8 (7.83 for 10-6 mol/L), which is the theoretical (thermodynamically calculated) value for the stability of Zn2+. This is important for the biodegradability of the alloy.
*and the Authors give a good disscusion in last paragraph of the disscusion section (lines 449-455)
Specific comments (typographic and grammar errors)
Abstract
Line 14: materials instead of material
Line 16: considerable biocompatibility instead of a considerable biocompatibility
Line 18: add a comma before “whereas”.
Line 23: qualifying instead of qualifiying
also a few errors in the use of articles and commas in the abstract.
Introduction
Line 27: Orthopedics is a non-British variant. The manuscript starts in British English, so Orthopaedics should be used; “are” instead of “is” should be used in the sentence.
Line 37: a comma should be added before including.
Line 40: applications instead of application
Line 46: article “the” instead of the article “a”
Line 97: dot instead of a comma.
Line 102: microscope instead of “micorscope”.
Lines 324, 325 and 327: Square mm should be shown in the exponent. Check for the entire manuscript!

They are given together with the general remarks.
Author Response
Dear Reviewer 1,
Thank you very much taking the time to review our manuscript.
English corrected throughout the manuscript
General Comments
A comparison of tensile strength between casted and extruded samples is missing. For the comprehensive material characterization given in the paper, samples' tensile strength and hardness should be included.
In our study, all assessed samples were extruded. For better understanding, we changed the description in 2.1. The entire manufacturing procedure of the samples (described in 2.1.) was conducted by Limedion and the end product (discs measuring 6 mm in diameter and 1 mm in thickness) was given to us for the assessment.
The mechanical properities as well as the assessment of the tensile strenght was not objective of this study.
- Materials and Methods
2.1. Sample Preparation and Characterization
This part (2.1.) is a little confusing and not easy to follow. It is not clear the importance of the dimensions of the alloy samples obtained from the supplier (Limedion GmbH) since it was melted and cast. More important is to give the dimensions of the casted sample, and that information is missing. Both dimensions should be shown (without explanation for the first; otherwise, it needs to be clarified). Authors also can provide just dimensions after extrusion. However, in 2.1.3. diameter of the cast ingot was given (30 mm), which is better to be in 2.1.
In our study, all assessed samples were extruded. For better understanding, we changed the description in 2.1. The entire manufacturing procedure of the samples (described in 2.1.) was conducted by Limedion and the end product (discs measuring 6 mm in diameter and 1 mm in thickness) was given to us for the assessment.
The mechanical properities as well as the assessment of the tensile strenght was not objective of this study.
Line 92: It is not clear for the Ag content whether it is to producers' specifications or by analysis in the study.
The Ag content is producers´specifications as mentioned in line 91, 92 in section 2.1.
Line 95: elevated temperatures (for the extrusion process) is not a precise term. Please give a precise temperature (or temperatures) for the operation.
We added the temperature in line 95 in section 2.1.
Line 97: The phosphating procedure is not given!
We added the phosphating procedure in line 102-104 in section 2.1.
2.1.2. Low vacuum has been used. Is this information just given by mistake? If not, a short explanation for an unusual procedure should be shown.
Thank you for noticing this mistake. We changed the value according to the used usual procedure in line 127 in section 2.1.2.
- Results
3.1. Sample Characterization
3.1.1 Grain Size Determination
Line 266: Authors gave the same magnifications (as common practice), but here it is not beneficial. Figure (b) should be shown with 10x higher magnification, or part of the image should be enlarged; otherwise, the grains are not visible at all.
We adjusted the figures in 3.1.1.
Line 269: The statistical sample is small for the purpose (average grain size). Fortunately, it is given approximatively so it can stay*.
*Standards for the number of grains per mm2 (and mean grain size) and software from Leica for the purpose, of performing the task on hundreds and thousands of grains.
Line 294: Are these EDX spectra necessary? They illustrate differences between two samples, but if authors consider them significant, then a short explanation is needed (tables 1 and 2 seem sufficient).
We revised the EDX spectra in 3.1.3. and removed the zoomed in spectra to avoid confusion. For explanation, the spectra do not show any differences but are zoomed in on a different scale.
Line 300: Figures at 160x and 40868x magnifications (a, b, g and h) are not illustrative for this conclusion. If authors insist on them, it should be noted that it is better visible in figures c-d and e-f
We removed the figures at 160x and 40868x magnifications.
Lines 303 and 304: Equipment and the parameters of measurement have already been given in the experimental part, and there is no need to repeat.
We decided to keep the equipment and the parameters of the measurement in the caption to provide the informations for a better reading flow. (Images must be understandale without reading the text and vice versa)
Lines 305-310 (the whole 3.1.4. part): ICP-OES just confirm the specifications from the producer of the alloy, and this could be moved to the experimental part. It looks superfluous (as a section in the Results part).
We decided to keep the ICP-OES to provide data about the alloys and their Ag content.
Line 315 (whole 3.1.5. part): Comment (short explanation) for the decrease in the concentration of metals is needed. For the zinc, it could be an extremely dealloying effect (selective dissolution of zinc) that leads to the higher silver concentration at the surface (but this need to be confirmed by EDS or some other analysis).
Although a highly sensitive graphite furnace AAS device was used, silver in chloride solutions is so poorly soluble that it was expected to be at the detection limit. On the other hand, in 0.1M chloride solution silver can exist as soluble in the form of AgCl2- ions in such extremely low concentrations (ng/ml, pg/L). Here one could even expect an increase in concentration with time, although that is more for a separate study and the use of ICP-MS. Here the results for silver are, in fact, inconclusive and maybe it can be stressed. Additionally, an increase in pH also can be the cause for the decrease in Zn2+ ion concentration, but here the buffer is used.
Thank you. We discussed this matter in line 429-435 in section 4.1.
Line 327: Are values permuted here (15 ± 19 and 14 ± 22)? (the deviations are greater than the values)
No, the values are not permuted. The number of found cells in the Live/Dead assay varied greatly.
Lines 366 and 375: Captions for Figures 9 and 10 are given incorrectly. Figure 9 has a caption for Fig. 10 and vice versa.
Thank you. The Captions have been corrected
Line 380: (just a positive comment, do not change anything, it is for the disscusion part*). Important result! This is, fortunately, under a pH of ~8 (7.83 for 10-6 mol/L), which is the theoretical (thermodynamically calculated) value for the stability of Zn2+. This is important for the biodegradability of the alloy.
*and the Authors give a good disscusion in last paragraph of the disscusion section (lines 449-455)
Specific comments (typographic and grammar errors)
Abstract
Line 14: materials instead of material
Corrected
Line 16: considerable biocompatibility instead of a considerable biocompatibility
Corrected
Line 18: add a comma before “whereas”.
Corrected
Line 23: qualifying instead of qualifiying
also a few errors in the use of articles and commas in the abstract.
Corrected
Introduction
Line 27: Orthopedics is a non-British variant. The manuscript starts in British English, so Orthopaedics should be used; “are” instead of “is” should be used in the sentence.
Corrected
Line 37: a comma should be added before including.
Corrected
Line 40: applications instead of application
Corrected
Line 46: article “the” instead of the article “a”
Corrected
Line 97: dot instead of a comma.
Corrected
Line 102: microscope instead of “micorscope”.
Corrected
Lines 324, 325 and 327: Square mm should be shown in the exponent. Check for the entire manuscript!
Corrected
Reviewer 2 Report
Dear Authors,
The manuscript presents a significant understanding of the subject matter, and I believe the examination seems interesting for the researchers. Connecting the methodology employed and the sample’s data collected leads to a reliable and meaningful result. The findings have the potential to contribute significantly to the field and develop further discussion among researchers. However, a few revisions for improving the manuscript are recommended. Therefore I do suggest a Major revision.
1. The manuscript should be reported by passive sentences, e.g. lines 16, 70, and so on. The authors are recommended to check all the sentences including the Abstract.
2. The number of titles in some parts should be corrected, e.g. in lines 279 and 296.
3. It seems necessary to review the article for spelling mistakes, e.g. in line 23, the word qualifying is written as “qualifiying”.
4. It seems necessary to revise the article from the point of view of grammar. e.g. in line 16, the word considerable is written as “a considerable”.
5. In lines 33 to 35, it seems that there is no need to refer to 8 articles to raise a question.
6. In lines 297-300, the reader will understand better if the numbers of the pictures are mentioned in the text.
7. Authors should check all the acronyms. They should be introduced as it appears in the context e.g. “DPBS”.
8. Since the study is about zinc and its alloys, the authors should emphasize on Zinc’s studies and metals in the Introduction section. The author should better use and cite these articles. https://doi.org/10.1016/j.jmbbm.2023.105785orhttps://doi.org/10.1016/j.icheatmasstransfer.2022.106067
9. Why did NOT the authors perform MTT assay test?
10. Since the study is about bio-absorbable implants, why a degradation test like PBS is NOT performed?
Minor editing of English language edit required.
Author Response
Dear Reviewer 2,
Thank you very much taking the time to review our manuscript.
English corrected throughout the manuscript.
- The manuscript should be reported by passive sentences, e.g. lines 16, 70, and so on. The authors are recommended to check all the sentences including the Abstract.
Corrected
- The number of titles in some parts should be corrected, e.g. in lines 279 and 296.
Corrected
- It seems necessary to review the article for spelling mistakes, e.g. in line 23, the word qualifying is written as “qualifiying”.
Corrected
- It seems necessary to revise the article from the point of view of grammar. e.g. in line 16, the word considerable is written as “a considerable”.
Corrected
- In lines 33 to 35, it seems that there is no need to refer to 8 articles to raise a question.
Thank you, we decided to refer to just the most important articles.
- In lines 297-300, the reader will understand better if the numbers of the pictures are mentioned in the text.
Corrected
- Authors should check all the acronyms. They should be introduced as it appears in the context e.g. “DPBS”.
Corrected
- Since the study is about zinc and its alloys, the authors should emphasize on Zinc’s studies and metals in the Introduction section. The author should better use and cite these articles. https://doi.org/10.1016/j.jmbbm.2023.105785orhttps://doi.org/10.1016/j.icheatmasstransfer.2022.106067
We cited the paper „The effects of atomic percentage and size of Zinc nanoparticles, and atomic porosity on thermal and mechanical properties of reinforced calcium phosphate cement by molecular dynamics simulation“. Unfortunately, the other paper could not be cited.
- Why did NOT the authors perform MTT assay test?
The WST assay is very similar to the MTT assay and assesses the cell viability and proliferation rate. Both assays use the reduction equivalents NADH and NADPH.
- Since the study is about bio-absorbable implants, why a degradation test like PBS is NOT performed?
The degradation test (AAS) was performed according to ISO standard 10993-15:2019-11.
Round 2
Reviewer 1 Report
No further comments. The paper has been improved enough to be published.
Reviewer 2 Report
Thank you for the revised version.
Minor editing of English language required